# Biomarkers for Outcome in Metastatic Melanoma in First Line Treatment with Immune Checkpoint Inhibitors

**DOI:** 10.3390/biomedicines11030749

**Published:** 2023-03-01

**Authors:** Tanja Mesti, Cvetka Grašič Kuhar, Janja Ocvirk

**Affiliations:** 1Institute of Oncology Ljubljana, Zaloška 2, 1000 Ljubljana, Slovenia; 2Faculty of Medicine, University of Ljubljana, Korytkova Ulica 2, 1000 Ljubljana, Slovenia

**Keywords:** immune checkpoint inhibitors, metastatic melanoma, predictive and prognostic biomarkers, immune-related adverse events, immune-inflammation parameters

## Abstract

Introduction: A high proportion of metastatic melanoma patients do not respond to immune checkpoint inhibitors (ICI), and until now, no validated biomarkers for response and survival have been known. Methods: We performed a retrospective analysis of outcomes in patients with metastatic melanoma treated with first-line ICI at the Institute of Oncology Ljubljana from January 2018 to December 2020. The immune-related adverse events (irAEs) and serum immune-inflammation parameters (neutrophil-to-lymphocyte ratio (NLR), platelet-to-lymphocyte ratio (LR), systemic immune-inflammation index (SII) and pan-immune-inflammation value (PIV)) were analyzed as potential biomarkers for response and survival. Survival rates were calculated using the Kaplan–Meier method and then compared with the log-rank test. Multivariate regression Cox analysis was used to determine independent prognostic factors for progression-free survival (PFS) and overall survival (OS). Results: Median follow-up was 22.5 months. The estimated median progression-free survival (PFS) was 15 months (95% CI 3.3–26.2). The two-year survival rate (OS) was 66.6%. Among 129 treated patients, 24 (18.6%) achieved complete response, 28 (21.7%) achieved partial response, 26 (20.2%) had stable disease and 51 (39.5%) patients experienced a progressive disease. There was a higher response rate in patients with irAEs (*p* < 0.001) and high NLR before the second cycle of ICI (*p* = 0.052). Independent prognostic factors for PFS were irAE (HR 0.41 (95% CI 0.23–0.71)), SII before the first cycle of ICI (HR 1.94 (95% CI 1.09–3.45)) and PLR before the second cycle of ICI (HR 1.71 (95% CI 1.03–2.83)). The only independent prognostic factor for OS was SII before the first cycle of ICI (HR 2.60 (95% CI 0.91–7.50)). Conclusions: Patients with high pre-treatment levels of SII had a higher risk of progression and death; however, patients with irAEs in the high-SII group might respond well to ICI. Patients who develop irAEs and have high NLRs before the second ICI application have higher rates of CR and PR, which implicates their use as early biomarkers for responsiveness to ICI.

## 1. Introduction

The annual incidence of malignant melanoma in Europe varies between 3 and 5 people per 100,000 in Mediterranean countries and between 12 and 35 people per 100,000 in Nordic countries [1]. In Slovenia, the average annual melanoma incidence rate in the period from 2014 to 2018 was 25.6 for females and 29.5 for males per 100,000. These numbers increased compared to the average annual melanoma incidence rate from 2009 to 2013 (24.3 for females and 25 for males per 100,000), and is estimated to increase further to 28 women and 34 men per 100,000 (95% prediction interval) in the year 2021. That makes Slovenia one of the European countries with the highest annual incidence of malignant melanoma. Approximately 78% of Slovenian patients with melanoma initially present with localized disease, which is mostly due to broad public education; 19% present with regional disease; and 3% with distant metastatic disease [2]. All Slovenian melanoma patients diagnosed with stage III and IV are treated with systemic therapy at the Institute of Oncology Ljubljana. The programmed death 1 (PD1) inhibitors pembrolizumab and nivolumab are the main inhibitors used, as well as anti-cytotoxic T-lymphocyte-associated protein 4 (CTLA-4) antibodies in combination with nivolumab, in accordance with the Slovenian national guidelines and in line with the international guidelines of the European Society for Medical Oncology and National Comprehensive Cancer Network for the treatment of melanoma [3,4,5]. Immune checkpoint inhibitors (ICI) against CTLA-4 and PD1 have initiated a breakthrough in the treatment and improved prognosis of patients with metastatic melanoma. The survival of patients treated with ICI increased from the expected survival time of less than 12 months to at least 40 months [6,7,8,9]. The first ICI approved for the treatment of metastatic malignant melanoma was ipilimumab, an anti-CTLA-4 antibody which is associated with a median OS of 11.4 months (95% CI, 10.7 to 12.1 months) and a 3-year survival rate of 22% (95% CI, 20% to 24%) [6]. Significantly higher survival rates were achieved with the ICIs pembrolizumab and nivolumab, directed against PD1, with a median survival of approximately 40 months and an estimated 5-year OS of 41% in treatment-naïve patients for pembrolizumab and 44% for nivolumab [6,7,8,9]. Ipilimumab in combination with nivolumab results in a further improvement of the 5-year OS rate to 52% and the median survival to more than 60 months [6]. The high efficacy of ICIs used as mono-therapy, which is even higher if used as doublets (ipilimumab and nivolumab), is connected with a broad spectrum of immune-related adverse events (irAE), such as immune-related skin toxicity, pneumonitis, thyroid dysfunction and other endocrinopathies, hepatitis and renal dysfunction [6,7,8,9]. Unlike BRAF (Raf murine sarcoma viral oncogene homolog B1) and MEK (mitogen-activated protein kinase) inhibitors, which inhibit the transmission of signals along the Ras/Raf/MEK signaling pathway, the mechanism of action of ICI is much more complex and, therefore, unpredictable. More than two-thirds of patients experience at least one irAE when treated with ICI. In approximately 10% of treated patients, the response to immunotherapy manifests as pseudoprogression, where there is a mismatch between the clinical and radiological response and the actual response, which represents a transiently violent inflammatory response [10]. In up to 30% of patients, ICI treatment induces hyperprogression, where cancer progresses even more rapidly than expected for unknown reasons. Half of the patients will not respond at all [10,11]. The response to ICI, pseudoprogression, hyperprogression and the occurrence of irAEs involves a variety of yet unknown mechanisms. As of now, there are no biopathological or clinical biomarkers for predicting ICI efficacy. Recent data are focusing on blood inflammatory parameters and their possible role as biomarkers for immunotherapy responses [12,13].

As previously published by our research team, we found a significant positive correlation between irAEs and the response rate to ICI. Patients that developed irAEs also had significantly better progression-free survival (PFS) [14]. Herein, we present the results of our further evaluation of the possible association of the immune-inflammation indexes, calculated from peripheral blood cells, with immune-related adverse events (irAEs) and survival outcomes in metastatic melanoma patients treated with ICI. We used a larger sample, covering patients for three consecutive years.

## 2. Materials and Methods

A retrospective analysis of patients with metastatic melanoma treated with ICI from January 2018 to December 2020 at the Institute of Oncology Ljubljana was performed. The study inclusion criteria were: metastatic melanoma patients undergoing first-line treatment with ICI, radiographic evaluations of immunotherapy treatment, having an Eastern Cooperative Oncology Group (ECOG) performance status of 0–3 and either without or with inactive autoimmune disease. Exclusion criteria were: 2nd or further line of treatment, active autoimmune disease, bad performance status (PS 4) and a lack of a radiologic evaluation of the treatment. Patients and tumor characteristics, laboratory parameters (lactate dehydrogenase (LDH), S 100 protein, leukocytes, neutrophils, basophils, eosinophils, lymphocytes and monocytes), molecular parameters (BRAF V600 and NRAS (neuroblastoma Ras gene)), irAEs, treatment responses and events such as relapse and death were collected from medical records.

The study was approved by the National Medical Ethics Committee (approval number: 0120-342/2020/5) and was carried out according to the Declaration of Helsinki.

The ICI was applied as an intravenous infusion at the recommended dose of 200 mg every 3 weeks for pembrolizumab and either 240 mg every 2 weeks or 480 mg every 4 weeks for nivolumab. Nivolumab was used in combination with ipilimumab as follows: 1 mg/kg nivolumab in combination with 3 mg/kg ipilimumab, administered intravenously every 3 weeks for the first 4 doses, followed by a second phase with nivolumab monotherapy, applied as described above.

The immune Response Evaluation Criteria in Solid Tumors (iRECIST) criteria were used to evaluate the tumor response [15]. The irAEs were evaluated by a clinician based on the findings of laboratory tests, clinical examinations and imaging studies, and were graded according to the National Cancer Institute Common Terminology Criteria for Adverse Events, version 5.0. [16]. Treatment interruption was defined as a time interval during which an irAE was assessed by the oncologist at a grade (CTCAE v 5.0) indicating that the ICI should be temporarily discontinued until the irAE is reversed. Treatment discontinuation was defined as a time point at which irAE was assessed at a grade (CTCAE v 5.0) indicating that the ICI should be terminated.

The systemic immune-inflammation markers were assessed from the peripheral blood platelet (P), neutrophil (N), monocyte (M) and lymphocyte (L) counts in cells/L [17,18,19]. The neutrophil-to-lymphocyte ratio (NLR) was calculated as N/L; the platelet-to-lymphocyte ratio (PLR) as P/L, the pan-immune-inflammation value (PIV) as (N × P × M)/L) and the systemic immune-inflammation value (SII) as P × N/L. We evaluated NLR, PLR, PIV and SII at baseline before the start of ICI therapy and before the second cycle of ICI. Cut-off values for low and high values were set based on data from the literature or median values were set at ≥2 for NLR, ≥180 for PLR, ≥390 for PIV and ≥730 for SII [17,18].

The OS was defined as the time from the date of the first ICI administration to the date of death from any cause. The PFS was defined as the time from the date of the beginning of ICI to the date of disease progression or death from any cause.

The characteristics of patients were categorically presented as frequencies and pro-portions. Age was presented as median and range. Pearson’s chi-square test was used for statistical comparisons for categorical data, and the unpaired Student’s *t*-test was used for comparing ages between groups. In the case of expected parameter values of <5 in >20% of cells, Fisher’s exact test was used to facilitate the analysis of smaller population sizes. A *p*-value ≤ 0.05 was considered statistically significant. Spearman rho was used for the calculation of the correlation between ordinal determinants.

The survival analyses were performed using the Kaplan–Meier method and com-pared by log-rank test. The prognostic significance of the variable of interest (age, ECOG performance status, melanoma type, metastatic site, LDH, comorbidities, irAE, response rate and immune-inflammatory parameters (NLR, PLR, PIV and SII) was calculated using the Cox proportional hazards regression model, and expressed as hazard ratio (HR) and 95% confidence interval (CI). All the variables showing a *p* ≤ 0.05 in the univariate models were included in the multivariable model. The variables showing a *p* ≤ 0.05 in the multivariable models were considered to be independent prognostic factors. All statistical analyses were performed using SPSS v.28.0 (IBM Corp., Armonk, NY, USA).

## 3. Results

### 3.1. Patients and Treatment

From January 2018 to December 2020, 311 patients were treated with ICI for malignant melanoma at the Institute of Oncology Ljubljana. Of these, 129 patients fulfilled the inclusion criteria for the study (Figure 1).

Patients (all Caucasian) were treated with ICI as the first-line treatment for metastatic melanoma. The median age of patients was 66.2 (30.1–84.5) years, and 61.3% were males. In total, 75% of patients had primary skin melanoma; 47.3% were metastatic to skin or soft tissue, including muscle and/or nonregional lymph nodes (stage M1a); and 17% had central nervous metastases. More than half (56.6%) of patients had concomitant diseases, the most prevalent of which was arterial hypertension (39.5%), and 7.8% had a history of an autoimmune disease (one patient had sarcoidosis; others had thyroid autoimmune disease). BRAF mutations were present in 23.3% and N-RAS in 6.2% of patients. Over three-quarters of patients (76.7%) were treated with pembrolizumab, and 37.2% of patients developed irAE. Patients’ baseline clinical and pathological characteristics, as well as the type of ICI treatment, are summarized in Table 1.

We compared clinical, pathological and systemic immune-inflammatory markers between the group of patients with occurrence irAE and the group without irAE (Table 2). Patients in the group with irAE were more often treated with ipilimumab + nivolumab. They had also higher systemic inflammation indices (SII) before the first cycle of ICI and a higher frequency of BRAF mutations.

### 3.2. Response Rates and Survival Outcomes

The median follow-up time was 22.5 months. The median PFS was 15 months (95% CI 3.3–26.6). In the group of patients without irAE, the median PFS was 9.3 months, compared with 32.8 months in the group with irAE (*p* = 0.01). Numbers in bold are statistically significant.

Univariate and multivariate analyses of possible factors affecting PFS are presented in Table 3.

In our cohort, it was revealed that patients with irAE had more than halvedthe hazards of progression (HR 0.41) that their counterparts without irAE did (Figure 2A). However, patients with high SII indices before the first cycle of ICI, and those with high PLR before the second cycle of ICI, had a risk of progression almost two times higher than the corresponding patients with low levels of SII and PLR (HR 1.94 and 1.71, respectively) (Figure 2B,C).

In the whole cohort, 1-year OS was 80% and 2-year OS was 66.6%. Median OS was not reached. An analysis of possible prognostic factors and their effects on OS (univariate and multivariate Cox analysis) is presented in Table 3. The only independent prognostic factor for OS was revealed to be SII before the first cycle of ICI (*p* = 0.003; Figure 3). Patients with high SIIs had hazards of death 2.6 times higher than patients with low SII. In patients with low SII, 2-year OS was 77%, compared to 55% in patients with high SIIs (*p* < 0.003; Figure 3).

### 3.3. Immune-Related Adverse Events and Immune-Inflammation Parameters

A total of 48 (37.2%) patients developed irAEs; 16 patients (11.7%) developed dermatitis; 10 patients (7.8%) thyroiditis, pneumonitis and/or colitis; 9 (6.3%) hepatitis; and in two patients (1.6%) other organs were affected (Appendix A). There were no deaths caused by irAEs, although 31 patients (24%) had treatment interruption and 18 patients (14%) had permanent treatment discontinuation due to irAEs. The median time to treatment interruption was 3 months (95% CI 1.6–4.5), and the median time to treatment discontinuation was the same (95% CI 0–6.4). The majority of treatment interruptions and discontinuations happened in the first 6 months of treatment with ICI (Appendix A).

In total, 24 (18.6%) patients achieved a complete response, 28 (21.7%) achieved a partial response, 26 (20.2%) had stable disease and 51 (39.5%) experienced progressive disease. The response to ICI was significantly associated with the occurrence of irAEs (*p* < 0.001) (Appendix A) and high NLR before the second cycle of ICI (*p* = 0.052). Patients with irAEs and high NLR before the second ICI had higher rates of complete and partial responses.

The subgroup of patients with high SIIs before the ICI, but who developed irAEs due the ICI treatment, had better PFS (Figure 4).

## 4. Discussion

We present data regarding possible new clinical biomarkers for predicting ICI efficacy in metastatic melanoma treatment-naïve patients. According to our data analysis, high pre-treatment levels of SII were connected with a high risk of progression and death. Subgroup analysis of patients with high SII according to irAEs revealed that patients with irAEs had higher PFS in comparison with those who did not. Patients who developed irAEs and had high NLR before the second ICI application were the best responders to ICI. The median PFS was 15 months (95% CI 3.3–26.6), and median OS was not reached. This corresponds with data from already-published studies. In Keynote 006 patients with metastatic melanoma treated with pembrolizumab, the median PFS was 9.7 months (95% CI, 5.8–12.0), and the median OS was not reached after a follow-up period of 48 months [20]. In CheckMate 067, the median PFS was 11.5 months (95% CI, 8.7 to 19.3) in the nivolumab-plus-ipilimumab group, 6.9 months (95% CI, 5.1 to 10.2) in the nivolumab group and 2.9 months (95% CI, 2.8 to 3.2) in the ipilimumab group, and the median OS was more than 60.0 months (median not reached) in the nivolumab and ipilimumab groups and 36.9 months in the nivolumab group, as compared with 19.9 months in the ipilimumab group (HR with nivolumab and ipilimumab vs. ipilimumab, 0.52; HR with nivolumab vs. ipilimumab, 0.63) [7]. Grade 3–4 irAEs occurred in 17% of the patients, and 1 patient died from treatment-related sepsis in Keynote 006 [20]. The correlation of irAEs and pan-immune-inflammation values with survival was not initially assessed for the metastatic patients in these two studies, but in 2020, Eggermont et al. published the results of a prospective study’s sub-analysis of pembrolizumab vs. placebo in high-risk stage III melanoma patients treated with pembrolizumab. This revealed a positive correlation of irAEs with longer recurrence-free survival [21]. In the same year, we published our retrospective analysis results regarding Slovenian metastatic melanoma patients treated with ICI, showing a higher survival probability of more than 80% in patients with irAEs vs. less than 60% in patients without irAEs [14]. The meta-analysis of irAEs in patients with different cancers and their correlation with the treatment efficacy, which included 52 papers comprising 9156 patients and pooled data analysis, demonstrated a greater and statistically significant probability of achieving an objective tumor response in patients with irAEs compared to those without (OR 3.91). Patients who developed irAEs presented prolonged PFS (HR 0.54) and OS (HR 0.51) rates. This refers mostly to NSCLC and melanoma patients treated with ICI, regardless of the grade of the irAE or discontinuation of the treatment [22]. Our study’s research data suggest the same notion, that irAE could be used as a potential biomarker for higher response rates and lower hazards of progression and death in patients treated with ICI.

The importance of systemic inflammation in tumor development, progression and metastasis has been proven [23]. Pro-tumorigenic cytokines secreted by neutrophils and platelets (vascular endothelial growth factor, tumor necrosis factor-α, interleukin-10, etc.) contribute to cancer progression. On the other hand, monocytes and lymphocytes have anti-tumoral effects, as they increase the immune response against the tumor [24]. Recently, systemic immune-inflammation prognosis scores, including PIV, NLR, PLR, monocyte-to-lymphocyte ratio (MLR) and SII, have been reported to be of prognostic value in many malignancy types, including melanoma [18,23,24,25,26,27,28,29,30,31,32,33,34,35,36,37,38,39]. The PIV is a scoring system that includes all immune-inflammatory cells in the peripheral blood count (neutrophils, platelets, monocytes and lymphocytes), and has proven to be a useful prognostic biomarker in some malignancies, such as colon and breast cancers. It has also been reported to be a strong predictor of outcomes in microsatellite instability–high metastatic colorectal cancer patients receiving immunotherapy [35,36,39]. SII represents a promising biomarker in cancers, such as hepatocellular, pancreatic, gastric, esophageal and small and non-small cell lung cancers [18,25,26,27,28,29,30,31,32,33,34,35,36,37,38,39].

In a metastatic colorectal setting, in patients with high microsatellite instability cancers treated with checkpoint inhibitors, both PIV and SII correlate with the PFS and OS, according to pooled data from the Valentino and Tribe trials. Patients with high PIV had 1.66-times-higher hazards of progression and two-times-higher hazards of death compared to patients with low PIV [13]. In metastatic melanoma patients treated with ICI, the data are not consistent enough to make reliable conclusions. Namely, based on Susok et al.’s prospective study, which included an analysis of 62 patients treated with immunotherapy for unresectable stage III and IV melanoma, PIV and SII did not seem to be significant predictors for clinical outcome. However, according to Fuca and al.’s retrospective study analysis of 228 metastatic melanoma patients treated with checkpoint inhibitors, a high baseline PIV was independently associated with lower OS (adjusted HR: 2.06) and PFS (adjusted HR 1.56). High PIV was also associated with primary resistance to immunotherapy (odds ratio (OR): 3.98) [12,35].

According to our data analysis, high pre-treatment SII had a 1.94-times-higher risk of progression and 2.6-times-higher risk of death compared with patients with low SII, which was similar to the results of the colorectal cancer study [13]. A subgroup analysis of patients with high SII according to irAE revealed that patients with irAEs had higher PFS in comparison with those who did not (*p* = 0.002) (Figure 4). Patients with high PLR before the 2nd ICI application had 1.71-times-higher risks of progression. Regarding the response rate to ICI, high NLR before the 2nd ICI application significantly correlated with better response, and those patients had higher CR and PR. This is the main contribution of our study, made in a time where there are no conclusive data regarding the potential role of irAEs and immune-inflammation parameters as biomarkers for ICI efficacy; thus, our research adds valuable data to this topic. First of all, the development of irAEs in patients with metastatic melanoma treated with ICI as a first-line treatment can be used as a potential biomarker to determine the subgroup which will likely achieve a good response and which has a better PFS rate. The other important message of our study is the potential use of initially high SII at presentation, as well as high PLR before the second ICI application, as early biomarkers for unresponsiveness to ICI. However, even if patients had an initially high SII but developed irAEs, they showed longer PFS rates than those without irAEs. Patients that developed irAEs and had high NLR before the second ICI application had higher rates of CR and PR.

The strength of our study is that we provide the results for the response and survival rates of patients treated at a single, wide, national and comprehensive cancer center. We strictly followed the inclusion criteria, regularly evaluated treatment efficacy according to iRECIST and used accurate vitality data. The limitations of our study include its retrospective study design.

## 5. Conclusions

In a time in which there are no conclusive data on the potential role of irAEs and immune-inflammation parameters as biomarkers for ICI efficacy in the treatment of metastatic melanoma patients, our research study analysis carries important messages. Patients with high pre-treatment levels of SII have higher risk of progression and death; however, patients with irAEs in the high-SII group might be good responders to ICIs. High PLR before the second cycle suggests a higher hazard of progression. Furthermore, patients who develop irAEs and have high NLR before the second ICI application have higher rates of CR and PR, which implicates their usefulness as early biomarkers for responsiveness to ICI.

## Figures and Tables

**Figure 1 biomedicines-11-00749-f001:**
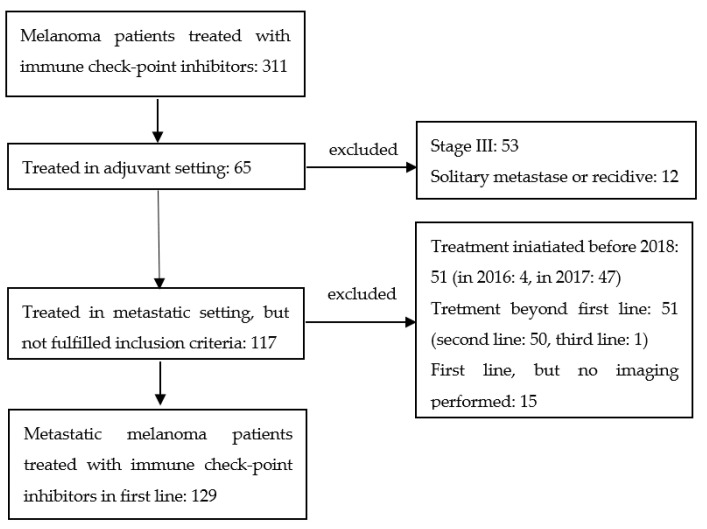
Consort diagram of patients in the study.

**Figure 2 biomedicines-11-00749-f002:**
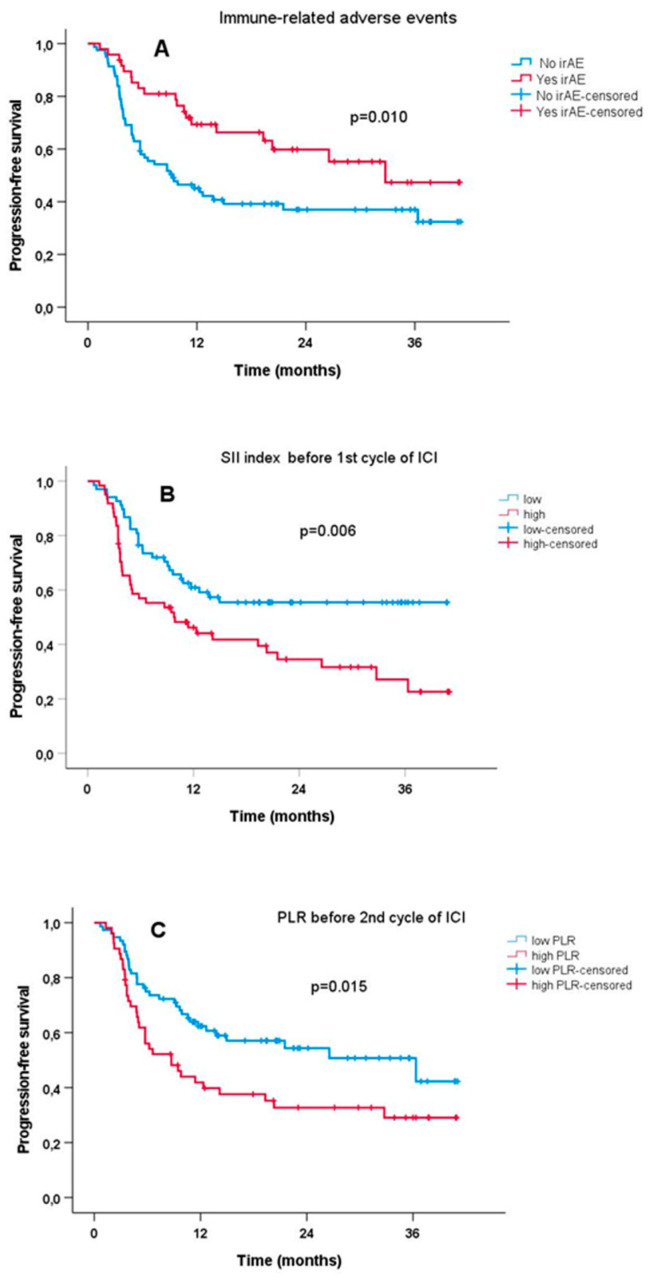
Kaplan–Meier curves for progression-free survival (PFS) in metastatic melanoma treatment-naïve patients treated with checkpoint inhibitors (ICI), according to occurrence of immune-related adverse events (irAE) (**A**), systemic immune-inflammation index (SII) (**B**) and platelet-to-lymphocyte ratio (PLR) (**C**).

**Figure 3 biomedicines-11-00749-f003:**
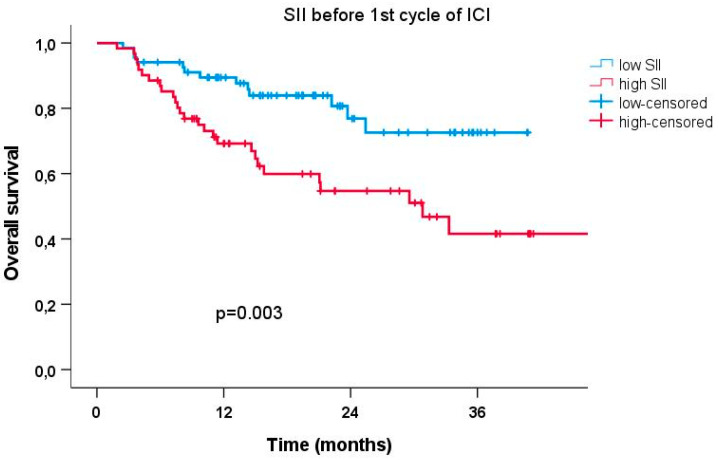
Kaplan–Meier curve for overall survival (OS) in metastatic melanoma treatment-naïve patients treated with checkpoint inhibitors, with regard to systemic immune-inflammation index (SII).

**Figure 4 biomedicines-11-00749-f004:**
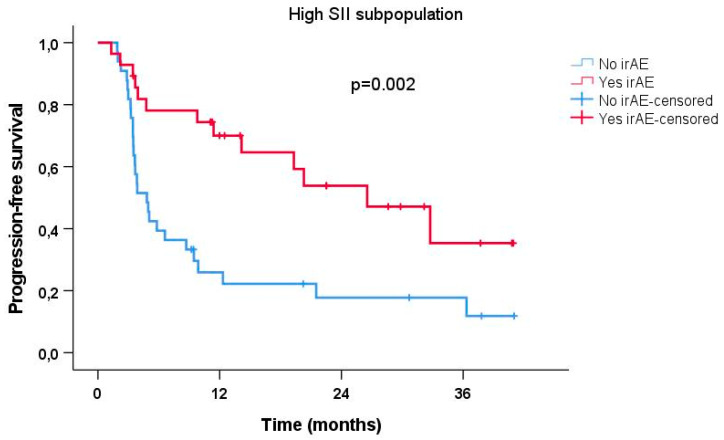
Kaplan–Meier curves for progression-free survival (PFS) in metastatic melanoma treatment-naïve patients treated with checkpoint inhibitors (ICI) according to occurrence of immune-related adverse events (irAEs) in correlation with systemic immune-inflammation index (SII).

**Table 1 biomedicines-11-00749-t001:** Baseline clinical and pathological characteristics, type of immune checkpoint inhibitor (ICI) treatment and frequency of immune-related adverse effects (irAE). M1a—skin, soft tissue, nonregional lymph node; M1b—lung; M1c—other visceral sites; M1d—the central nervous system. BRAF: gene that encodes protein B-Raf.

Characteristics	Value	Number (%)
**Total**		129 (100)
**Age (years): median (range)**	66.2 (30.1–84.5)	
**Gender**	Male	84 (61.3)
	Female	53 (38.7)
**Melanoma type**	Skin	97 (75.2)
	Uveal	11 (8.5)
	Mucosal	2 (1.6)
	Unknown origin	19 (14.7)
**Metastatic site**	M1a	61 (47.3)
	M1b	20 (25.5)
	M1c	26 (20.2)
	M1d	22 (17.1)
**BRAF gene mutation**	BRAF wild type	74 (57.4)
	V600E mutation	20 (15.5)
	V600K mutation	8 (6.2)
	V600K and V600M mutation	1 (0.8)
	V600R	1 (0.8)
	Testing not performed	25 (19.4)
**Comorbidity**	No comorbidity	56 (43.4)
	Arterial hypertension	51 (39.5)
	Diabetes	13 (10.1)
	Pulmonary disease	5 (3.9)
	Autoimmune disease	10 (7.8)
	Other disease	46 (35.7)
**Type of ICI treatment**	Pembrolizumab	99 (76.7)
	Nivolumab	14 (10.9)
	Nivolumab and Ipilimumab	16 (12.4)
**irAE**	No	81 (62.8)
	Yes	48 (37.2)

**Table 2 biomedicines-11-00749-t002:** Comparison of clinicopathological and systemic inflammatory characteristics between groups of patients who either developed immune-related adverse effects (irAE) or did not (statistically significant marked bold). M1a—distant metastasis to skin or soft tissue, including muscle and/or nonregional lymph nodes; M1b—lung; M1c—other visceral sites; M1d—the central nervous system. LDH—lactate dehydrogenase, ECOG—Eastern Cooperative Oncology Group, PS—performance status, PIV—pan-inflammation value, SII—systemic immune-inflammation index, PLR—platelet-to-lymphocyte ratio, NLR—neutrophil-to-lymphocyte ratio, ICI—immune checkpoint inhibitor.

Characteristics		Group without irAE N (%)	Group with irAE N (%)	All Patients N (%)	*p*-Value
**Number**		81 (62.8)	48 (37.2)	129 (100)	
**Age (years)**	<65 years =>65 years	45 (55.6) 36 (44.4)	26 (54.2) 22 (45.8)	71 (55) 58 (45)	0.878
**Melanoma type**	Skin	59 (72.8)	38 (79.2)	97 (75.2)	0.452
	Uveal	6 (7.4)	5 (10.4)	11 (8.5)	
	Mucosal	2 (2.5)	0 (0)	2 (1.6)	
	Unknown origin	14 (17.3)	5 (10.4)	19 (14.7)	
**Metastatic site**	M1a	34 (42)	27 (56.3)	61 (47.39)	0.275
	M1b	14 (17.3)	5 (10.4)	19 (14.7)	
	M1c	15 (18.5)	10 (20.8)	25 (19.4)	
	M1d	18 (22.2)	6 (12.5)	24 (18.6)	
**ECOG PS**	0	39 (48.1)	20 (41.7)	59 (45.7)	0.492
	1	29 (35.8)	23 (47.9)	52 (403)	
	2	12 (14.8)	5 (10.4)	17 (13.2)	
	3	1 (1.2)	0 (0)	1 (0.8)	
**Concomitant diseases**	No	36 (44.4)	20 (41.7)	56 (43.4)	0.758
	Yes	45 (55.6)	28 (58.3)	73 (56.6)	0.758
**BRAF status mutation ***	No	52 (81.2)	22 (55)	74 (71.2)	**0.004**
	Yes	12 (18.8)	18 (45)	30 (28.8)	
**LDH**	normal	58 (71.6)	40 (83.3)	98 (76)	0.132
	elevated	23 (28.4)	8 (16.79)	31 (34)	
**Treatment**	Pembrolizumab	68 (84)	31 (64.6)	99 (76.7)	**0.004**
	Nivolumab	9 (11.1)	5 (10.4)	14 (10.9)	
	Nivolumab and ipilimumab	4 (4.9)	12 (25)	16 (12.4)	
**Immune-inflammatory indexes**					
**NLR before 1st cycle of CPI**	low	25 (30.9)	9 (18.8)	34 (26.4)	0.131
	high	56 (69.1)	39 (81.2)	95 (73.69)	
**NLR before 2nd cycle of ICI**	low	20 (24.7)	10 (20.8)	30 (23.3)	0.616
	high	27 (33.3)	38 (79.2)	99 (76.7)	
**PLR before 1st cycle of ICI**	Low	54 (66.7)	26 (54.2)	80 (62)	0.157
	High	27 (33.3)	22 (45.8)	49 (32)	
**PLR before 2nd cycle of ICI**	Low	51 (63)	25 (52.1)	76 (58.9)	0.225
	High	30 (37)	23 (47.9)	53 (41.1)	
**PIV before 1st cycle of ICI**	Low	47 (58)	20 (41.7)	67 (51.9)	0.072
	High	34 (42)	28 (58.3)	62 (48.1)	
**PIV before 2nd cycle of ICI**	Low	37 (45.7)	24 (50)	61 (47.3)	0.635
	High	44 (54.3)	24 (50)	68 (52.7)	
**SII before 1st cycle of ICI**	Low	48 (59.3)	20 (41.7)	68 (52.7)	**0.053**
	High	33 (40.7)	28 (58.3)	61 (47.3)	
**SII before 2nd cycle of ICI**	High	48 (59.3)	25 (52.1)	73 (56.6)	0.427
	High	33 (40.7)	23 (47.9)	56 (43.4)	

* Data for patients with available data.

**Table 3 biomedicines-11-00749-t003:** Univariate and multivariate Cox regression analyses of prognostic factors associated with prognosis. PFS—progression free survival, OS—overall survival, irAE—patients with metastatic melanoma who developed immune-related side effects due to immunotherapy, NirAE—patients with metastatic melanoma who did not develop immune-related side effects due to immunotherapy, CR—complete response, PR—partial response, SD—stabile disease, PD—progressive disease, PIV—pan-inflammation value, SII—systemic immune-inflammation index, PLR—platelet-to-lymphocyte ratio, NLR—neutrophil-to-lymphocyte ratio, LDH—lactate dehydogenase, >4.31 microkat/L. Numbers in bold are statistically significant.

Factors	Overall Survival				Progression–Free Survival			
	Univariate Analysis		Multivariate Analysis		Univariate Analysis		Multivariate Analysis	
	HR (95% CI)	*p*	HR (95% CI)	*p*	HR (95% CI)	*p*	HR (95% CI)	*p*
Gender								
female vs. male	0.82 (0.44–1.54)	0.535			0.91 (0.57–1.46)	0.695		
Age								
≥65 years vs. <65 years	1.62 (0.86–3.03)	0.134			0.91 (0.57–1.46)	0.695		
Melanoma type								
uveal vs. skin	2.09 (0.80–5.41)	0.131			1.50 (0.68–3.32)	0.316		
mucosal vs. skin	1.75 (0.24–12.95)	0.584			2.82 (0.68–11.6)	0.152		
unknown vs. skin	0.90 (0.35–2.32)	0.82			1.16 (0.60–2.23)	0.658		
Location of metastases								
m1b vs. m1a	1.35 (0.48–3.78)	0.572	0.66 (0.16–2.80)	0.57	0.91 (0.43–1.92)	0.811		
m1c vs. m1a	2.79 (1.29–6.04)	**0.009**	1.68 (0.65–4.29)	0.282	1.45 (0.79–2.65)	0.231		
m1d vs. m1a	2.26 (0.96–5.31)	0.062	1.86 (0.55–6.27)	0.317	1.28 (0.67–2.46)	0.455		
Ecog ps								
1 vs. 0	1.75 (0.84–3.68)	0.137	0.98 (0.39–2.45)	0.963	0.99 (0.59–1.66)	0.973		
≥2 vs. 0	3.76 (1.66–8.53)	0.002	1.13 (0.30–4.29)	0.243	1.39 (0.68–2.83)	0.185		
Comorbidities								
yes vs. no	0.65 (0.45–1.20)	0.167			0.62 (0.39–0.99)	**0.047**	0.64 (0.40–1.03)	0.065
Ldh								
elevated vs. normal	3.13 (1.65–5.92)	**<0.001**	1.30 (0.42–4.01)	0.643	1.61 (0.94–2.77)	0.082		
S100								
elevated vs. normal	2.42 (1.30–4.50)	**0.005**	2.61 (0.91–7.50)	0.074	1.30 (0.80–2.11)	0.298		
Braf mutation								
yes vs. no	0.30 (0.09–0.98)	**0.047**	0.28 (0.07–1.10)	0.067	/			
Type of treatment (ici)								
nivolumab vs. pembrolizumab	0.69 (0.21–2.56)	0.691			1.35 (0.66–2.75)	0.415		
nivolumab+ipilimumab vs. pembrolizumab	0.92 (0.28–3.07)	0.923			1.25 (0.59–2.66)	0.558		
Irae								
yes vs. no	0.44 (0.21–0.93)	**0.031**	0.39 (0.14–1.05)	0.062	0.51 (0.30–0.86)	**0.012**	0.41 (0.23–0.71)	**0.002**
Nlr before 1st cycle of ici								
high vs. low	2.16 (0.91–5.15)	0.082			1.24 (0.71–2.16)	0.457		
Nlr before 2nd cycle of ici								
high vs. low	1.42 (0.63–3.22)	0.398			1.82 (0.95–3.47)	0.069		
Plr before 1st cycle of ici								
high vs. low	1.63 (0.88–3.03)	0.122			1.48 (0.92–2.39)	0.104		
Plr before 2nd cycle of ici								
high vs. low	1.58 (0.85–2.98)	0.149			1.78 (1.11–2.86)	**0.017**	1.71 (1.03–2.83)	**0.038**
Piv before 1st cycle of ici								
high vs. low	1.78 (0.94–3.35)	**0.075**			1.31 (0.82–2.10)	0.266		
Piv before 2nd cycle of ici								
high vs. low	1.34 (0.71–2.51)	0.364			1.10 (1.05–2.75)	**0.033**	1.08 (0.61–1.91)	0.802
Sii before 1st cycle of ici								
high vs. low	2.64 (1.36–5.12)	**0.004**	2.60 (0.91–7.50)	**0.026**	1.92 (1.19–3.10)	**0.008**	1.94 (1.09–3.45)	**0.025**
Sii before 2nd cycle of ici								
high vs. low	1.59 (0.85–2.95)	0.146			1.57 (0.98–2.53)	0.06		

## Data Availability

The research data for this study are not publicly available on legal and ethical grounds. Regulation (EU) 2016/679—General Data Protection Regulation (GDPR) protection of natural persons with regard to the processing of personal data and the free movement of such data. Further enquiries can be directed to the corresponding author.

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
