# Peer review of "Biomarkers for Outcome in Metastatic Melanoma in First Line Treatment with Immune Checkpoint Inhibitors"

_biomedicines, 2023, doi:10.3390/biomedicines11030749_

Round 1

Reviewer 1 Report (Previous Reviewer 2)

In response to critiques, I believe the manuscript has been significantly improved and now possibly warrants publication in Biomolecules. The only concern I still have is the English language which requires a professional revision.

Author Response

Thanks, we revised accordingly.

Reviewer 2 Report (Previous Reviewer 4)

The manuscript looks good after extensive revision, I would suggest making figures more meaningful visually. As some of them are still not clear visually or the use of colors is not homogenous 

Author Response

Dear reviewer

Please tell us, did you meant for the tables or figures and please can you give us precise suggestions.

Kind regards,

Authors

This manuscript is a resubmission of an earlier submission. The following is a list of the peer review reports and author responses from that submission.

Round 1

Reviewer 1 Report

This is an exceptionally good manuscript, and I am very glad the authors submit this to Biomedicines. In general, I find the manuscript very clearly written. The article highlights important data about possible new clinical biomarkers for predicting ICI efficacy. The authors found that patients with high SII before the first cycle of ICI were associated with lower responses to ICI therapy. On the other hand, patients with irAE, had a higher response rate to ICI. High SII was found also an independent unfavourable prognostic factor for PFS and OS.

Only a few minors:

Please introduce the Aberration the first time coming up in the manuscript, e.g. SII, HR.

Please note the uppercase letters in the table and unify the figures in tone, format, and font.

Please add the title for the X axis and the unit on the Y axis for Progression-free survival. Titles of some figs need to be revised too. 

Reviewer 2 Report

This manuscript requires extensive reviewing from a biostatistician and a cancer immunologist. Many information has anecdotal value and English language is impaired. I found also many flaws, which overall make the mnuscript unpublishable in the present format. I am just enumerating some big weaknesses below.

Progressive disease (PD) is NOT a prognostic factors for PFS in cancer. PD is part of defining PFS (lines 19-20).

5-year overall survival rate of 20 months (line 56)…. 5-year overall survival rate of approximately 40 months (line 58). 5-y OS is a percentage of live people at 5 years post-diagnosis or post-starting treatment, by definition.

Some critical assumptions are not supported by literature, such “The interaction between inflammatory pro-tumour populations (i.e. neutrophils, platelets and monocytes) and anti-cancer immune populations (i.e. lymphocytes)”. There are N1 and N2, M1 and M2, many lymphocytes subpopulations are pro-tumor e.g., (Th2, Treg, Th17), many lymphocytes remain naïve with no clear function in controlling tumor immunity since no priming will occur, etc.
Comorbidities are discussed as yes or not, but not shown in detail.

Results description do not correspond the values in the tables. For example:

-“Two thirds of patients had primary skin melanoma” (line 151), but in the table 1 the corresponding value is 75.2%.

-Section 3.4 results do not fit at all with the data from its corresponding figure 4.

Overall, many sections from Discussion do not integrate with the topic.

Minor comments:

There are many repetitions in the text for the same information…some hilarious, like “intravenous infusion intravenously “ (line 97)

Many information in the paper requires corresponding references.

Many abbreviations are defined repetitively.

Please spell irAE (line 12) and pts (line 27) in the abstract. Please also remove pts abbreviation from the whole manuscript. This is not a common abbreviation.

Please spell SII in line 19 instead 24.

Please define N-RAS (line 157).

Conjunction till is not appropriate for a scientific paper. Please use until instead of till.

Reviewer 3 Report

In this retrospective study the Authors report on their experience on the use of immune check point inhibitors for Stage III and IV melanoma and potential prediction biomarkers. 

The Authors have put a significant effort in the preparation of this manuscript but some details need attention: 

1. Abstract. Explain all abbreviations. LIne 23: What ist less progressive disease.

Materials and Methods:

2. Lines 10-11 . "The interaction between....(NLR) was also performed. Maybe the inclusion of a small Talbe here to analyse the calcuations performed will be useful to the general reader. 

Results

The results should be rewritten and presented in a more clearway. For example Page 6, Lines 17-18. "Response to ICI and SII before the 1st cycle of ICI revealed...". This is not clear as is also not clear the sentence that follows. 

Likewise "Median OS was not reached". Obviously means the expecte overall survive, yet it should be made more reader friendly. 

Although there was no difference in the PFS between Pembrolizumab and Nivolumab plus/minus Ipilimumab, it would be interested to report on their respective effects on blood inflammation markers. 

When the Authors state in Figure 3 Treatment interruption and treatment discontinuation, they should define it in M&M section.

Figure 5, Figure 6. Is there a statistical significance?

Discussion

It should focus on the present findings and not on the response rates observed in othes studies. Therefore the discussion should be thoroughly modified excluding long referene to the efficacy of ICI in other studies. 

Reviewer 4 Report

The manuscript, Biomarkers for outcome in metastatic melanoma in first line treatment with immune checkpoint inhibitors is interesting, focusing the light on the blood inflammatory parameters and its possible role as biomarker for immunotherapy response. I would like to make some general comments on that,

1- The scope and rationale of the study must be described meaningfully in abstract and in introduction

2- At some instances, the statements need literature support e.g., "Till today there are no bio-pathological or clinical biomarkers for predicting ICI efficacy"

3- Figure 6. Progression free survival (PFS) is little blurred to understand.